# “Radiological Grading” for Preoperative Assessment of Central Cartilaginous Tumors

**DOI:** 10.3390/cancers18020193

**Published:** 2026-01-07

**Authors:** Shinji Miwa, Katsuhiro Hayashi, Takashi Higuchi, Hirotaka Yonezawa, Sei Morinaga, Yohei Asano, Satoru Demura

**Affiliations:** Department of Orthopedic Surgery, Graduate School of Medical Sciences, Kanazawa University, 13-1 Takara-machi, Kanazawa 920-8640, Japan; hysk@med.kanazawa-u.ac.jp (K.H.);

**Keywords:** cartilaginous tumor, chondrosarcoma, histological grade, imaging

## Abstract

In cartilaginous tumors, surgeons sometimes experience discrepancies in the histological grade between the preoperative biopsy and resected tumor specimen. For cartilaginous tumors, new diagnostic tools or methods for the prediction of histological grades are required to determine appropriate surgical procedures for each patient. Several radiological findings have been reported to be useful in differentiating benign cartilaginous tumors, atypical cartilaginous tumors/grade 1 chondrosarcomas, and high-grade chondrosarcomas. Furthermore, recent studies have shown the high accuracy of radiological scoring systems that integrate several radiological findings to predict the histological grades of cartilaginous tumors. Radiomics, which converts features in radiological images into quantitative data, enables the comprehensive analysis of cartilaginous tumors. This review article discusses radiological findings, integrated radiological scoring systems, and radiomics-based predictions of histological grades in cartilaginous tumors.

## 1. Introduction

Chondrosarcomas, which produce the chondroid matrix, account for 20–27% of primary malignant bone tumors [1]. Chondrosarcomas are classified into conventional (grades 1–3), clear-cell, periosteal, dedifferentiated, and mesenchymal subtypes [2]. In the WHO 2013 classification, “atypical cartilaginous tumor (ACT)” was first described as a cartilaginous tumor with locally aggressive behavior arising in the appendicular bone [3]. In the WHO 2020 classification, ACT and grade 1 chondrosarcoma were clearly defined as a locally aggressive, hyaline cartilage-producing neoplasm [4]. Tumor in the appendicular bone was defined as ACT (intermediate), whereas tumors in the axial bone were defined as grade 1 chondrosarcoma (malignant). Among the subtypes of cartilaginous tumors, dedifferentiated and mesenchymal chondrosarcomas have a highly malignant behavior and a high incidence of distant metastases. Because most chondrosarcomas are resistant to chemotherapy and radiation therapy, tumor resection with appropriate surgical margins is important. Intralesional resection is acceptable for ACT/grade 1 chondrosarcomas, whereas tumor resection with an appropriate surgical margin is required for high-grade chondrosarcomas [5]. Therefore, preoperative radiological and histological assessments are necessary to resect cartilaginous tumors appropriately.

Although biopsy is the standard method for preoperative diagnosis, high incidences of discordance between preoperative and postoperative histological grades have been reported in cartilaginous tumors [6,7,8]. Roitman et al. investigated the concordance between needle biopsy and the postoperative histological grading of central chondrosarcomas [6]. In this study, the concordance between biopsy and final histological diagnosis of chondrosarcoma was 83% in the long bones, whereas the concordance of the diagnosis was only 36% in the pelvis. Concordance in the differentiation of ACT/grade 1 chondrosarcomas and high-grade chondrosarcomas in the long bones and pelvis was 90% and 67%, respectively. Tsuda et al. investigated the correlation between the preoperative histological grade and final diagnosis in patients with peripheral chondrosarcomas of the pelvis [7]. In the study, only 15 out of 41 patients (37%) showed concordance in histological grade between preoperative biopsy and final diagnoses of resected tumor specimens. In another study, the diagnostic accuracies of frozen and permanent section diagnoses of biopsy specimens were investigated by comparing them with the final diagnosis of the excised tumor specimen in patients with bone tumors [8]. In cartilaginous tumors, the accuracies of frozen and permanent section diagnoses for the differentiation of benign cartilaginous tumors and ACT/chondrosarcomas were 84% and 93%, respectively, which were lower than those of other types of bone tumors. Based on these reports, the accuracy of the current histological diagnosis of chondrogenic tumors remains unsatisfactory. Therefore, novel diagnostic methods and modalities with high accuracy are required to determine appropriate treatment plans for cartilaginous tumors. This review article discusses various radiological findings correlated with histological grades, comprehensive radiological scoring systems, and radiomics-based predictions of the histological grades of cartilaginous tumors. This review included studies based on the following criteria: (1) studies on the correlation between radiological findings and histological grades in cartilaginous tumors, (2) clearly defined radiological findings, scoring systems, or radiomics/machine learning-based prediction models, (3) inclusion of patients with histological diagnoses of cartilaginous tumors, and (4) publication in peer-reviewed international journals.

## 2. Radiological Findings to Predict Histological Grade

To determine treatment planning, including surgical margins, histological diagnosis using biopsy specimens is necessary for musculoskeletal tumors. However, surgeons sometimes experience discordance in histological diagnoses based on biopsy specimens of cartilaginous tumors, and diagnostic discordance may cause inappropriate surgery. Therefore, careful radiological assessment using radiography, computed tomography (CT), magnetic resonance imaging (MRI), and/or nuclear medicine is required before surgery. In previous studies, the usefulness of several radiological findings in radiography, CT, MRI, and nuclear medicine has been reported for the prediction of histological grade in cartilaginous tumors [9,10,11,12,13,14,15,16,17]. The characteristic radiological findings of ACT/grade 1 chondrosarcoma and high-grade chondrosarcoma for differentiation from enchondroma are shown in Table 1. Because ACT and grade 1 chondrosarcoma are histologically identical, these tumors were classified into the same group. Murphey et al. reported that radiological features, including periosteal reaction on radiography, deep endosteal scalloping (greater than two-thirds of cortical thickness), cortical destruction, extraskeletal mass on CT or MRI, and increased uptake of radionuclides (greater than the anterior iliac crest) on bone scintigraphy, strongly suggested the diagnosis of chondrosarcoma [18]. Geirnaerdt et al. investigated the diagnostic ability of radiological features, including margins, sclerotic rim, contour, osteolysis, intralesional sclerosis, calcifications, cortical thinning, expansion, scalloping, periosteal reaction, and soft tissue extension, to differentiate between enchondromas and chondrosarcomas [13]. In this study, ill-defined margins were observed in 67% of chondrosarcomas and 37% of enchondromas (*p* = 0.004), and multilobulated lesions were observed in 72% of chondrosarcomas and 43% of enchondromas (*p* = 0.009). Alfaro et al. investigated the clinical and radiological findings to distinguish enchondromas from ACT/grade 1 chondrosarcoma of the pelvis [19]. In this study, age (*p* = 0.006), endosteal scalloping (*p* = 0.039), and tumor size (*p* = 0.005) were significantly associated with ACT/grade 1 chondrosarcoma, whereas no significant difference in pain was observed (*p* = 0.553). Choi et al. investigated MRI findings to distinguish between enchondromas and ACT/grade 1 chondrosarcoma [9]. Among MRI findings, intermediate signal intensity on T1-weighted images (72% ACT/grade 1 chondrosarcoma vs. 25% enchondromas), multilocular lesions on contrast-enhanced T1-weighted images (83% vs. 44%), cortical destruction (33% vs. 0%), extraskeletal mass (28% vs. 0%), abnormal signal intensity in adjacent bone marrow and soft tissues (22% vs. 0%), and involvement in the epiphyseal or flat bone (56% vs. 19%) were frequently observed in ACT/grade 1 chondrosarcoma. Douis et al. investigated the role of conventional and contrast-enhanced MRI in differentiating enchondroma and chondrosarcoma [20]. In this study, high incidences of large tumor, endosteal scalloping > 2/3, cortical destruction, bone expansion, and soft tissue masses were observed in grade 1 chondrosarcomas, whereas contrast-enhanced MRI was not useful in differentiating enchondromas from grade 1 chondrosarcomas. This finding suggests that these radiological findings can be used to differentiate between enchondromas and grade 1 chondrosarcoma.

Jo et al. investigated the role of thallium-201 (^201^Tl) and technetium-99m pentavalent dimercaptosuccinic acid (^99m^Tc-DMSA) scintigraphy in grading cartilaginous tumors [21]. This study included 64 patients (21 enchondromas and 43 chondrosarcomas) who underwent both ^201^Tl- and ^99m^Tc-DMSA scintigraphy. Positivity for scintigraphy was defined as greater uptake than that of the background. ^201^Tl uptake was more frequent in grade 1 chondrosarcomas (odds ratio = 7.92) than in enchondromas, and ^201^Tl uptake was significantly associated with the histological grades. Although ^99m^Tc-DMSA uptake was more frequent in chondrosarcomas than in enchondromas (odds ratio = 4.75), ^99m^Tc-DMSA uptake was not significantly associated with the histological grade of chondrosarcomas. They concluded that the combined use of ^201^Tl and ^99m^Tc-DMSA may increase diagnostic confidence in cartilaginous tumors. Annovazzi et al. investigated the ability of ^18^F-fluorodeoxyglucose (^18^F-FDG) positron emission tomography (PET)/CT to predict histological grades in 95 patients with cartilaginous tumors [16]. The optimal cutoff value of maximum standardized uptake (SUVmax) was 2.6 to differentiate between enchondroma and ACT/grade 1 chondrosarcoma (specificity 0.86, sensitivity 0.68), 3.7 to differentiate between ACT/grade 1 chondrosarcoma and grade 2–3 chondrosarcoma (specificity 0.84, sensitivity 0.83), and 7.7 to differentiate between grade 2–3 and dedifferentiated chondrosarcoma (specificity 0.90, sensitivity 0.92). Based on these results, ^18^F-FDG PET/CT seems reliable for differentiating grade 2–3 and dedifferentiated chondrosarcomas but is unsuitable for differentiating between benign cartilaginous tumors and ACT/grade 1 chondrosarcoma.

There are studies showing the usefulness of several radiological findings for differentiation between ACT/grade 1 chondrosarcomas and high-grade chondrosarcomas (Table 2). Kaya et al. investigated the correlation between Tl-201 uptake and histological grade in 23 patients with cartilaginous tumors [10]. In this study, increased uptake of Tl-201 was observed in all patients with grade 3 chondrosarcoma and 2 of 5 patients with grade 2 chondrosarcoma, whereas 3 of 5 grade 2 chondrosarcomas and all 7 grade 1 chondrosarcomas showed no Tl-201 uptake in the tumor region. This result indicates that Tl-201 scintigraphy may be useful to differentiate between ACT/grade 1 chondrosarcoma and high-grade chondrosarcoma. Yoo et al. investigated MRI imaging features for differentiation between ACT/grade 1 chondrosarcomas and high-grade chondrosarcomas in 42 patients with chondrosarcomas [22]. High signal intensity in a central area on T1-weighted images (*p* < 0.01), absence of entrapped fat within the tumor (*p* < 0.01), absence of internal lobular structures (*p* < 0.01), soft tissue formation (*p* < 0.01), and large central non-enhancing areas on gadolinium-enhancing areas (*p* < 0.01) were more frequently observed in high-grade chondrosarcomas. Multivariate analysis showed that the absence of entrapped fat within the tumor and soft tissue formation were independently associated with high-grade chondrosarcomas. Douis et al. investigated the association between findings on MRI and histological grades in 179 patients with atypical cartilaginous tumors/chondrosarcomas [23]. In this study, multivariate analysis showed that active periostitis (*p* = 0.001), tumor length (*p* < 0.001), soft tissue mass (*p* < 0.001), and bone expansion (*p* = 0.001) were significantly associated with high-grade chondrosarcomas.

**Table 1 cancers-18-00193-t001:** Clinical and radiological features for differentiating atypical lipomatous tumor/chondrosarcoma from enchondroma.

Author	Patients	Modality	Findings	Ref.
Geirnaerdt	Enchondroma (35) and chondrosarcoma (43)	X-ray	Ill-defined margin and lobulated contour	[13]
Murphy	Enchondroma (92) and chondrosarcoma (95)	X-ray	Cortical remodeling, cortical thickening, cortical destruction, pathologic fracture, periosteal reaction, and soft-tissue extension	[18]
		CT	cortical destruction, periosteal reaction, and soft-tissue extension	
		MRI	Cortical destruction and soft-tissue extension	
		Bone scintigraphy	Uptake greater than in the anterior iliac crest	
Brenner	Chondrosarcoma (31)	FDG-PET	Mean SUV in grade 1, 2, and 3 chondrosarcomas were 3.4 ± 1.6, 5.4 ± 3.1, and 7.1 ± 2.6, respectively	[24]
Higuchi	Enchondroma (3) and chondrosarcoma (19)	^201^Tl scintigraphy	Only mesenchymal and dedifferentiated chondrosarcoma showed obvious ^201^Tl uptake	[11]
Choi	Enchondroma (16) and chondrosarcoma (18)	MRI	Findings correlating ACT/grade 1 chondrosarcoma: intermediate signal on T1-weighted images, multilocular appearance on contrast-enhanced T1-weighted images, cortical destruction, a soft tissue mass, adjacent bone marrow and soft tissue abnormal signal, and involvement of the epiphysis or flat bone	[9]
De Coninck	Enchondroma (75) and chondrosarcoma (31)	Dynamic contrast-enhanced MRI	A two-fold more relative enhancement compared with muscle (100% sensitivity and 63.3% specificity)	[25]
Crim	Enchondroma (32) and chondrosarcoma (12)	X-ray	Size, endosteal scalloping, cortical breakthrough, and bone expansion	[26]
		MRI	Size, cortical breakthrough, large areas of enhancement by gadolinium, and soft mass	
Alfaro	Enchondroma (5) and ACT/grade 1 chondrosarcoma (16)	X-ray, CT, and MRI	Endosteal scalloping and tumor size	[19]
Douis	Enchondroma (27) and chondrosarcoma (23)	MRI	Tumor length, endosteal scalloping > 2/3, cortical destruction, bone expansion, and soft tissue mass	[20]
Jo	Enchondroma (21) and chondrosarcoma (43)	^201^Tl scintigraphy	Positivity for ^201^Tl uptake, defined as greater uptake compared with that of background, was more frequent in grade 1 chondrosarcoma	[21]
Annovazzi	Enchondroma (35) and chondrosarcoma (60)	FDG-PET	Cutoff value of SUVmax was 2.6 to differentiate between enchondroma and ACT/grade 1 chondrosarcoma, 3.7 to differentiate between ACT/grade 1 chondrosarcoma and grade 2–3 chondrosarcoma, and 7.7 to differentiate between ACT/grade 1–3 chondrosarcoma and dedifferentiated chondrosarcoma	[16]

**Table 2 cancers-18-00193-t002:** Radiological features for differentiating high-grade chondrosarcoma from low-grade chondrosarcoma.

Author	Patients	Modality	Finding	Ref.
Yoo	ACT/grade 1 chondrosarcoma (28) and high-grade chondrosarcoma (14)	MRI	Absence of entrapped fat and soft tissue mass formation	[22]
Douis	ACT/grade 1 chondrosarcoma (107), and high-grade chondrosarcoma (72)	MRI	Bone expansion, active periostitis, soft tissue mass, and tumor length	[23]
Kaya	Enchondroma (7) and chondrosarcoma (16)	^201^Tl scintigraphy	Increased ^201^Tl uptake greater than background was observed in grade 2–3 chondrosarcoma	[10]

Each modality provides complementary information, and a comprehensive assessment of the modalities may improve diagnostic accuracy and make it possible to determine the appropriate surgical procedure. However, radiological grading has several limitations, including interobserver variability, subjective interpretation, and modality-dependent differences. Radiological assessment using a single imaging modality may have low reliability, and a comprehensive and quantitative radiological assessment may prevent the misdiagnosis of grading in cartilaginous tumors.

## 3. Comprehensive Radiological Assessment to Predict Histological Grades

Although previous reports have shown a correlation between various radiological findings and the histological grades of cartilaginous tumors, the accuracy of these findings is thought to be limited. In contrast, previous reports suggest that comprehensive assessments using several radiological findings with quantitative assessment could enable the evaluation of the aggressiveness of cartilaginous tumors. Miwa et al. developed a radiological scoring system that combines radiography, CT, MRI, bone scintigraphy, and ^201^Tl scintigraphy [27]. In this study, periosteal reaction on radiography, scalloping and cortical defects on CT, extraskeletal mass, multilobular lesion, abnormal signal in adjacent tissue on MRI, and increased uptake in bone and thallium scans were significantly correlated with the final histological diagnoses. Based on these results, an overall radiological scoring system was developed by combining radiological findings from radiography, CT, MRI, bone scintigraphy, and ^201^Tl scans. Radiological findings were weighted according to their K values, with 1–3 points assigned to each finding for differentiation between enchondromas and ACTs/chondrosarcomas. The overall radiological score rate was calculated as a percentage of the total score relative to the maximum possible score. The mean overall radiological score rates were 11.6%, 62.1%, 64.7%, and 84.3% for enchondromas, ACT/grade 1 chondrosarcomas, grade 2 chondrosarcomas, and grade 3/dedifferentiated/mesenchymal chondrosarcomas, respectively. The overall radiological score rate showed a sensitivity of 87.5%, specificity of 88.9%, and accuracy of 88.2% in differentiating between enchondromas and ACT/grade 1, and high-grade chondrosarcoma in the validation cohort (kappa value = 0.764). Gundavda et al. investigated the utility of a radiological aggressiveness score that included nine radiological findings from radiography and MRI [28]. One point was assigned to each parameter, and the radiological aggressiveness score (RAS) was defined as the sum of the points. Among the nine parameters, five, including cortical erosion, periosteal reaction, pathological fracture, extraskeletal mass, and bone marrow edema, were significantly associated with a higher grade of cartilaginous tumors. Parameters with RAS score ≥ 4 had a 97.9% sensitivity and 90.5% specificity for predicting grade 2–3 chondrosarcoma (AUC = 0.972).

## 4. Radiomics-Based Prediction of Histological Grades

Radiomics, an imaging-based technique that converts many features in radiological images into quantitative high-dimensional data, enables the comprehensive analysis of whole tumor characteristics [29,30]. Gitto et al. investigated the diagnostic value of X-ray radiomics-based machine learning for differentiating ACT/grade 1 chondrosarcoma from high-grade chondrosarcoma [31]. In the study, 150 patients with ACT/grade 1 chondrosarcoma and high-grade chondrosarcomas were divided into training and test cohorts. Five radiomic features, including busyness, area, perimeter-to-area ratio, strength, and perimeter, passed dimensionality reduction. The accuracy, sensitivity, and specificity of the machine learning classifier were 80%, 83%, and 79%, respectively, in the internal test cohort and 80%, 89%, and 67%, respectively, in the external test cohort. No difference in diagnostic performance was observed between X-ray radiomics-based machine learning and radiologists. Li et al. developed a CT-based radiomics nomogram for the prediction of histological grade in chondrosarcoma [32]. In this study, 196 patients were divided into training (*n* = 139) and validation (*n* = 57) cohorts. Endosteal scalloping, tumor size, and periostitis were selected as parameters to construct the nomogram. Kaplan–Meier survival analysis showed that nomogram scores were significantly associated with recurrence-free survival, and patients with high nomogram scores experienced tumor recurrence 2.7 times more frequently than patients with low nomogram scores. Nie et al. developed a CT-based deep learning radiomics model to predict the clinical outcomes of chondrosarcoma [33]. In the study, 211 patients with chondrosarcomas (127 in the training cohort and 84 in the test cohort) were enrolled. ACT/grade 1 chondrosarcoma stratified using a deep learning radiomics model showed significantly better recurrence-free survival than high-grade chondrosarcomas. Yildirim et al. constructed machine learning models using CT-based radiomics analysis to differentiate ACT/grade 1 chondrosarcomas from enchondromas [34]. In the study, CT images of 30 enchondromas and 26 chondrosarcomas were retrospectively reviewed, and 107 radiomics features were obtained for each patient. Among the 107 features, the five most important features, including coarseness, 10th percentile, energy, total energy, and sphericity, were identified using an algorithm-based information gain. To differentiate chondrosarcoma from enchondroma, the area under the curve (AUC) values were 0.950 for classification using all features and 0.967 for classification using five features. This study demonstrated that CT-based radiomics analysis is effective in differentiating between ACT/grade 1 chondrosarcomas and enchondromas.

Studies have demonstrated the utility of MRI-based radiomics for predicting the histological grades of cartilaginous tumors [35,36]. Li et al. investigated the diagnostic performance of an MRI-based radiomics nomogram for the differentiation of ACT/grade 1 chondrosarcomas and high-grade chondrosarcomas [35]. This study enrolled 44 patients with ACT/grade 1 chondrosarcomas and 58 patients with high-grade chondrosarcomas who were divided into training and validation cohorts. Multivariate analysis showed that only bone marrow edema (OR = 0.29, *p* = 0.012) was an independent predictor of high-grade chondrosarcoma. An MRI-based radiomics nomogram was constructed by combining the clinicoradiological predictor and radiomics signature. The predictive performance of the nomogram was better than that of clinicoradiological factors and radiomics signature. Gitto et al. investigated the diagnostic performance of MRI radiomics-based machine learning for differentiating ACT/grade 1 chondrosarcoma from grade 2 chondrosarcomas [36]. In this study, 158 patients with cartilaginous bone tumors were divided into training (74 patients with ACT/grade 1 chondrosarcoma and 19 patients with grade 2 chondrosarcoma) and external test (45 patients with ACT/grade 1 chondrosarcoma and 20 patients with grade 2 chondrosarcoma) cohorts. After tuning the training cohort, the machine learning classifier had 92% accuracy for differentiating cartilaginous tumors in the test cohort. The accuracy in the identification of ACT/grade 1 chondrosarcoma and grade 2 chondrosarcoma were 98% and 80%, respectively, and the classifier showed no difference in accuracy compared to the radiologist.

One study showed the usefulness of single-photon emission CT (SPECT)/CT for the prediction of histological grades of cartilaginous tumors. Yoon et al. investigated the diagnostic value of Tc-99m hydroxymethylene diphosphonate (HDP) SPECT/CT radiomics parameters in differentiating ACT/grade 1 chondrosarcoma from enchondromas [37]. In the study, 49 patients with ACT/grade 1 chondrosarcoma and enchondromas were classified into training (*n* = 32) and test (*n* = 17) cohorts. LASSO regression analysis selected two radiomics features, including zone-length non-uniformity for zone (ZLNUGLZL) and coarseness for neighborhood gray-level difference (CoarsenessNGLDM). Multivariate analysis revealed that a higher ZLNUGLZL was an independent predictor of ACT/grade 1 chondrosarcoma. The sensitivity and specificity of ZLNUGLZL were 85.0% and 58.3%, respectively, in the training cohort, and 83.3% and 90.9%, respectively, in the test cohort. This study suggests that HDP SPECT/CT radiomics may be useful in differentiating between enchondromas and ACT/grade 1 chondrosarcoma.

Prediction models that use radiomics and machine learning allow the extraction of many quantitative features that reflect tumor aggressiveness. These techniques may enable the prediction of clinical outcomes, including metastasis and local recurrence. However, these prediction models, such as radiomics-based deep learning models, have limitations, including small sample size, retrospective study, heterogeneity of imaging protocols, class imbalance, overfitting risk, and limited external or prospective validation. Additionally, radiomics-based prediction models are sensitive to image acquisition and reconstruction parameters, which may affect reproducibility. Understanding these limitations in studies on outcome prediction models is required to avoid overinterpretation of the studies. To apply a radiomics-based prediction model in clinical practice, high reproducibility with technical standardization, including acquisition parameters, segmentation methods, feature harmonization, and cross-scanner variability, is demanded. Furthermore, advances in standardized imaging protocols, multicenter radiomics datasets, and explainable AI are needed to apply these techniques to clinical decision algorithms.

## 5. Future Perspectives and Clinical Implications

Enchondroma and ACT/grade 1 chondrosarcoma can be treated with curettage, whereas high-grade chondrosarcoma should be resected with a wide surgical margin. To determine the surgical procedure, histological diagnosis of the biopsy specimen is necessary. However, high incidences of discrepancies of histological grades between biopsy specimens and resected tumor specimens have been reported. Cartilaginous tumors commonly show histological variety and heterogeneity, and biopsy specimens usually include only a limited part of the tumor, resulting in a high rate of sampling failure of the aggressive portion of the tumor. Patients with sampling failure undergo inappropriate surgical treatment due to the misdiagnosis of the histological grades before surgeries. Based on recent studies on radiological prediction of histological grades, a concept of “radiological grading” is emerging as a significant decision-making tool in the treatment of cartilaginous tumors. Radiological findings, including tumor volume, cortical destruction, periosteal reaction, and extraskeletal masses, can reflect tumor activity and aggressiveness. Radiological grading may play an important role in the diagnosis of cartilaginous tumors in the future. Furthermore, integrated diagnosis of preoperative histological and radiological assessment may improve diagnostic accuracy, resulting in appropriate decision-making for surgical procedures.

Recently, hybrid imaging techniques such as PET-CT and multiparametric MRI have shown their advantages, including improvement of diagnostic accuracy and comprehensive analysis of functional and structural assessment. Yin et al. developed a clinical radiomics nomogram based on three-dimensional multiparametric MRI features and clinical characteristics to predict early recurrence of pelvic chondrosarcoma [38]. In this retrospective study, 103 patients were divided into training (*n* = 72) and validation (*n* = 31) cohorts. In the clinical radiomics nomogram, radiomics features, especially multisequence combined features, are more important than clinical characteristics. A clinical radiomics nomogram based on combined features (T1 + T2 + contrast-enhanced T1) and clinical characteristics showed an AUC of 0.891 and accuracy of 0.857 in the validation cohort. Li et al. developed and validated preoperative multiparametric MRI-based radiomics to differentiate skull base chordoma and chondrosarcoma [39]. In this retrospective study, 210 patients were divided into the training and validation cohorts. Among 1941 radiomic features acquired from T1, T2, and contrast-enhanced T1-weighted images, 11 discriminative features were selected. The multiparametric radiomic signature, which consisted of 11 selected features, had an AUC of 0.975 and 0.872 in the training and validation cohorts, respectively. Hybrid or multiparametric imaging approaches may improve the reliability of the assessment of radiological grading in cartilaginous tumors.

## 6. Conclusions

Recent studies have demonstrated high reproducibility in the prediction of histological grades of cartilaginous tumors using a comprehensive assessment of imaging features, radiomics, and machine learning, which are highly reliable and personalized approaches. Radiological grading, which combines radiological findings and radiomics-based quantitative assessments, may enable the improvement of surgical decision-making for cartilaginous tumors.

## Data Availability

No new data were created or analyzed in this study.

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
