# Peer review of "“Radiological Grading” for Preoperative Assessment of Central Cartilaginous Tumors"

_cancers, 2026, doi:10.3390/cancers18020193_

Round 1
Reviewer 1 Report
Comments and Suggestions for Authors
This is a comprehensive, informative, and well-written review article. I have two minor edits to suggest, and one non-actionable comment. I applaud the authors on their article.
Minor edits:
Table 2 is not referred to in the main text. It is common practice to mention each table or figure at least once in the main text—please add a reference in the main text to Table 2 for completeness.
Lines 165-166: These lines are a very close wording to a line in citation [25]. As a precaution I would advise the authors to reword so that there is no risk of text overlap with a previously published work (even if it is from their own previous publication).
Manuscript, lines 165-166: “Based on the K values of the radiological findings for differentiating between enchondromas and ACTs/chondrosarcomas…”
Miwa et al [25]: “Based on the K values of the radiological findings for differentiation between enchondroma and ACTs/chondrosarcoma…”
Non-actionable comment:
On radiomics: the performance of models may show optimism bias, as even with appropriate segregation of training/testing data sets, the published model is the survivor of what is likely several attempts at the research exploratory level. This can lead to an implicit conflation of training and testing data sets at the research-program/pre-publishing level. This is an intrinsic limitation to the radiomics field, and not an issue that is specific to this review article; however, it is appropriate for me as a reviewer to acknowledge it. The authors do state in lines 259-260 that "Furthermore, a standardized assessment method is required for applying radiological grading in clinical practice", which possibly implicitly covers my comment above. This is a non-actionable comment.
Author Response
Response to reviewer 1
This is a comprehensive, informative, and well-written review article. I have two minor edits to suggest, and one non-actionable comment. I applaud the authors on their article.
Minor edits:
Table 2 is not referred to in the main text. It is common practice to mention each table or figure at least once in the main text—please add a reference in the main text to Table 2 for completeness.
Response:
Thank you for your suggestion.
The following sentences were added to the manuscript.
There are studies showing usefulness of several radiological findings for differentiation between low-grade and high-grade chondrosarcomas. Kaya et al. investigated the correlation between Tl-201 uptake and histological grade in 23 patients with cartilaginous tumors. In this study, increased uptake of Tl-201 was observed in all patients with grade 3 chondrosarcoma and 2 of 5 patients with grade 2 chondrosarcoma, whereas 3 of 5 grade 2 chondrosarcomas and all of 7 grade 1 chondrosarcoma showed no Tl-201 uptake in the tumor region. This result indicate that Tl-201 scintigraphy may be useful to differentiate between low-grade chondrosarcoma and high-grade chondrosarcoma. Yoo et al. investigated MRI imaging features for differentiate between low-grade and high-grade chondrosarcomas in 42 patients with chondrosarcomas. High signal intensity in a central area on T1-weighted images (P < 0.01), absence of entrapped fat within the tumor (P < 0.01), absence of internal lobular structures (P < 0.01), soft tissue formation (P < 0.01), and large central non-enhancing areas on gadolinium-enhancing areas (P < 0.01) were more frequently observed in high-grade chondrosarcomas. Multivariate analysis showed that absence of entrapped fat within the tumor and soft tissue formation were independently associated with high-grade chondrosarcomas. Douis et al. investigated association between findings on MRI and histological grades in 179 patients with atypical cartilaginous tumors/chondrosarcomas. In this study, multivariate analysis showed that active periostitis (P = 0.001), tumor length (P < 0.001), soft tissue mass (P < 0.001), and bone expansion (P = 0.001) were significantly associated with high-grade chondrosarcomas.
Lines 165-166: These lines are a very close wording to a line in citation [25]. As a precaution I would advise the authors to reword so that there is no risk of text overlap with a previously published work (even if it is from their own previous publication).
Manuscript, lines 165-166: “Based on the K values of the radiological findings for differentiating between enchondromas and ACTs/chondrosarcomas…”
Miwa et al [25]: “Based on the K values of the radiological findings for differentiation between enchondroma and ACTs/chondrosarcoma…”
Response:
Thank you for your pointing this out.
The sentence was corrected as follows:
Radiological findings were weighted according to their K values, with 1–3 points assigned to each finding for differentiation between enchondromas and ACTs/chondrosarcomas. The overall radiological score rate was calculated as a percentage of the total score relative to the maximum possible score.
Non-actionable comment:
On radiomics: the performance of models may show optimism bias, as even with appropriate segregation of training/testing data sets, the published model is the survivor of what is likely several attempts at the research exploratory level. This can lead to an implicit conflation of training and testing data sets at the research-program/pre-publishing level. This is an intrinsic limitation to the radiomics field, and not an issue that is specific to this review article; however, it is appropriate for me as a reviewer to acknowledge it. The authors do state in lines 259-260 that "Furthermore, a standardized assessment method is required for applying radiological grading in clinical practice", which possibly implicitly covers my comment above. This is a non-actionable comment.
Reviewer 2 Report
Comments and Suggestions for Authors
Ref.: Manuscript Number: cancers-4046856: “Radiological grading”for preoperative assessment of central cartilaginous tumors.
Comments to authors
This review article discusses radiological findings, integrated radiological scoring systems, and radiomics-based predictions of histological grades in cartilaginous tumors.
The following remarks must be considered by authors.
1/ This article highlights that, given the limitations of biopsy in cartilaginous tumors, imaging is evolving from a purely descriptive role to a true decision-making tool, with radiological grading emerging as a key concept for the future.
2/ Although radiological grading and radiomics appear highly promising, their clinical applicability remains limited by the lack of standardized protocols and multicenter validation, which makes the reproducibility of diagnostic performance uncertain.
3/ What criteria did you use to select the included studies?
4/ How do you manage the heterogeneity of imaging protocols?
5/ Why can a global radiological assessment, in some cases, better reflect tumor aggressiveness than a preoperative biopsy?
6/ Do you think that radiomics could eventually replace biopsy in certain clinical situations?
7/ What are the current limitations of these new methods?
8/ Do you consider an integrated diagnostic approach combining histology, conventional imaging, and radiomics to be the most reliable strategy at present?
9/ What are the future challenges in this field?
Author Response
Response to reviewer 2
This review article discusses radiological findings, integrated radiological scoring systems, and radiomics-based predictions of histological grades in cartilaginous tumors.
The following remarks must be considered by authors.
1. This article highlights that, given the limitations of biopsy in cartilaginous tumors, imaging is evolving from a purely descriptive role to a true decision-making tool, with radiological grading emerging as a key concept for the future.
Response:
Thank you for your pointing this out.
The following sentence was added to the Future perspectives and clinical implications.
Based on the recent studies on radiological prediction of histological grades, a concept of "radiological grading" is emerging as significant decision-making tool in the treatment of cartilaginous tumors.
2. Although radiological grading and radiomics appear highly promising, their clinical applicability remains limited by the lack of standardized protocols and multicenter validation, which makes the reproducibility of diagnostic performance uncertain.
Response:
Thank you for your comment on the limitation. We agree that the lack of standardized protocols and multicenter validation limit the clinical application of the radiological grading and radiomics-based prediction models.
The following sentence was added to the section “Future perspectives and clinical implications.”
Furthermore, advances in standardized imaging protocols, multicenter radiomics datasets, and explainable AI are needed to apply these techniques into clinical decision algorithms.
3. What criteria did you use to select the included studies?
Response:
We appreciate this question.
Although methodology of systematic review was not applied, this review included studies based on the following criteria: (1) studies on correlation between radiological findings and histological grade in cartilaginous tumors, (2) clearly defined radiological findings, scoring systems, or radiomics/machine learning based prediction model, (3) inclusion of patients with histological diagnoses of cartilaginous tumors, (4) publication in peer-reviewed international journals.
4. How do you manage the heterogeneity of imaging protocols?
Response:
Thank you for your pointing this out.
We agree that heterogeneity of imaging protocols is a major problem in this field. In this review, we addressed heterogeneity by focusing on reproducible radiological findings, rather than imaging protocol-dependent technical parameters. In the "Future perspectives and clinical implication," needs for standardization of imaging protocols and external validation were emphasized.
5. Why can a global radiological assessment, in some cases, better reflect tumor aggressiveness than a preoperative biopsy?
Response:
Cartilaginous tumors have been known for histological variety and heterogeneity. Biopsy specimens include only a limited part of the tumor, and sampling of the aggressive portion of tumor is difficult. In contrast, radiological assessment can reflect tumor aggressiveness according to the tumor volume, metabolic activity, and surrounding structures, such as cortical destruction, periosteal reaction, and extraskeletal masses. Therefore, comprehensive radiological assessment may obtain important information of aggressiveness that are not fully represented in biopsy specimens.
6. Do you think that radiomics could eventually replace biopsy in certain clinical situations?
Response:
Currently, we do not believe that radiomics can replace biopsy in clinical practice. However, radiomics may take a complemental role of the diagnosis of cartilaginous tumors in clinical practice. Further validation and improved reproducibility of radiomics-based prediction models can play a more important role in clinical practice in the future.
7. What are the current limitations of these new methods?
Response:
The main limitations of the studies on radiomics-based prediction model include small sample sizes, retrospective study, heterogeneity of imaging protocols, and lack of external validation. Additionally, radiomics-based prediction models are sensitive to image acquisition and reconstruction parameters, which may affect reproducibility.
8. Do you consider an integrated diagnostic approach combining histology, conventional imaging, and radiomics to be the most reliable strategy at present?
Response:
Thank you for your question.
We believe that integrated diagnostic approach combining histological assessment, conventional radiological findings, and radiomics-based prediction is the most reliable method to predict the aggressiveness of cartilaginous tumors. Each modality provides complementary information, and comprehensive assessment of the modalities may improve diagnostic accuracy and make it possible to determine the appropriate surgical procedure. This concept is emphasized in the section of "Future perspectives and clinical implication."
9. What are the future challenges in this field?
Response:
Thank you for your question.
The following sentences were added to the future perspectives and clinical implication section.
To apply radiomics-based prediction model into clinical practice, high reproducibility with technical standardization including acquisition parameters, segmentation methods, feature harmonization, and cross-scanner variability. Furthermore, advances in standardized imaging protocols, multicenter radiomics datasets, and explainable AI are needed to apply these techniques into clinical decision algorithms.
Reviewer 3 Report
Comments and Suggestions for Authors
-
The manuscript should more clearly discuss the limitations of individual radiological findings (e.g., interobserver variability, modality-dependent differences) and how these affect the reliability of grade prediction.
-
The structure and content of Table 1 need revision: patient subgroups and grade categories should be defined consistently, and criteria for pooling or separating data between enchondroma, ACT, and different chondrosarcoma grades must be explicitly justified.
-
The radiomics sections should address technical standardization in much greater depth, including acquisition parameters, segmentation methods, feature harmonization, and cross-scanner variability, since these are critical for reproducibility.
-
The review should provide a more quantitative comparison of existing radiological scoring systems (e.g., consolidated accuracy, sensitivity, specificity, AUC) and clearly state in which clinical settings each is preferable or limited.
-
Important emerging imaging approaches, such as hybrid or multiparametric imaging (e.g., combined metabolic and structural techniques), are not covered and should be added to provide a comprehensive state-of-the-art overview.
-
The discussion of outcome prediction models (radiomics nomograms and deep learning models) needs a more critical appraisal of sample size, class imbalance, overfitting risk, and the limited external or prospective validation.
-
Terminology regarding ACT vs. grade 1 chondrosarcoma and WHO 2013 vs. 2020 classifications should be unified throughout the text, with a clear explanatory paragraph early in the introduction.
-
The suggestion that radiological grading might serve as an alternative to biopsy is currently too strong; it should be toned down and conditioned on future prospective, ideally multicenter, validation studies comparing combined histological and radiological approaches.
-
The future perspectives section should better define specific research priorities (e.g., standardized imaging protocols, multicenter radiomics datasets, explainable AI) and how they could be translated into clinical decision algorithms.
-
The manuscript would benefit from clearer integration of the narrative: for each modality and each model, explicitly link how the reported diagnostic or prognostic performance could change concrete surgical decisions or follow-up strategies.
Author Response
Response to reviewer 3
1. The manuscript should more clearly discuss the limitations of individual radiological findings (e.g., interobserver variability, modality-dependent differences) and how these affect the reliability of grade prediction.
Response:
Thank you for your recommendation.
The limitations of individual radiological findings, including interobserver variability, subjective interpretation, and modality-dependent differences, were added to the Future perspectives and clinical implications section. Furthermore, it was emphasized that radiological assessment by single imaging modality may have low reliability, and that comprehensive and quantitative radiological assessment may prevent the misdiagnosis of grading in cartilaginous tumors.
2. The structure and content of Table 1 need revision: patient subgroups and grade categories should be defined consistently, and criteria for pooling or separating data between enchondroma, ACT, and different chondrosarcoma grades must be explicitly justified.
Response:
Thank you for your suggestion.
Table 1 aimed to show characteristic radiological findings of ACT and chondrosarcoma for differentiation from enchondroma. To ensure consistent terminology and clearer definition of patient subgroups and histological grades, the tumors were classified into enchondroma, atypical cartilaginous tumor, and chondrosarcoma.
3. The radiomics sections should address technical standardization in much greater depth, including acquisition parameters, segmentation methods, feature harmonization, and cross-scanner variability, since these are critical for reproducibility.
Response:
I agree with your comments.
The following sentences were added to the Future perspectives and clinical implications section.
To apply radiomics-based prediction model into clinical practice, high reproducibility with technical standardization including acquisition parameters, segmentation methods, feature harmonization, and cross-scanner variability.
4. The review should provide a more quantitative comparison of existing radiological scoring systems (e.g., consolidated accuracy, sensitivity, specificity, AUC) and clearly state in which clinical settings each is preferable or limited.
Response:
Thank you for your suggestion.
The quantitative performance metrics (accuracy, sensitivity, specificity, and AUC) for radiological scoring systems were added to the manuscript.
One of the scoring systems aimed to differentiate enchondroma and ACT/chondrosarcoma, and another system aimed to differentiate enchondroma/ACT and chondrosarcoma. Therefore, it is difficult to compare the utility of the systems. The purposes of the scoring systems were emphasized in the manuscript.
5. Important emerging imaging approaches, such as hybrid or multiparametric imaging (e.g., combined metabolic and structural techniques), are not covered and should be added to provide a comprehensive state-of-the-art overview.
Response:
Thank you for your valuable suggestion.
The following sentences were added to Future perspectives and clinical implications section.
Recently, hybrid imaging techniques such as PET-CT and multiparametric MRI have shown their advantages including improvement of diagnostic accuracy and comprehensive analysis of functional and structural assessment.
Yin et al. developed a clinical radiomics nomograms based on three dimensional multiparametric MRI features and clinical characteristics to predict early recurrence of pelvic chondrosarcoma. In this retrospective study, 103 patients were divided into training (n = 72) and validation (n = 31) cohorts. In clinical radiomics nomogram, radiomics features, especially multisequence combined features, are more important than clinical characteristics. Clinical radiomics nomogram based on combined features (T1 + T2 + contrast-enhanced T1) and clinical characteristics showed an AUC of 0.891 and accuracy of 0.857 in the validation cohort.
Li et al. investigated developed and validated a preoperative multiparametric MRI-based radiomics to differentiate skull base chordoma and chondrosarcoma. In this retrospective study, 210 patients were divided into training and validation cohort. Among 1941 radiomic features acquired from T1, T2, and contrast-enhanced T1 weighted images, 11 discriminative features were selected. The multiparametric radiomic signature, consisted of 11 selected features, had AUC of 0.975 and 0.872 in the training and validation cohorts, respectively. Thes hybrid and multiparametric imaging approaches may improve the reliability of the assessment of radiological grading in cartilaginous tumors.
Refenreces:
Li L, Wang K, Ma X, Liu Z, Wang S, Du J, Tian K, Zhou X, Wei W, Sun K, Lin Y, Wu Z, Tian J. Radiomic analysis of multiparametric magnetic resonance imaging for differentiating skull base chordoma and chondrosarcoma. Eur J Radiol. 2019 Sep;118:81-87
Yin P, Mao N, Liu X, Sun C, Wang S, Chen L, Hong N. Can clinical radiomics nomogram based on 3D multiparametric MRI features and clinical characteristics estimate early recurrence of pelvic chondrosarcoma? J Magn Reson Imaging. 2020 Feb;51(2):435-445
6. The discussion of outcome prediction models (radiomics nomograms and deep learning models) needs a more critical appraisal of sample size, class imbalance, overfitting risk, and the limited external or prospective validation.
Response:
We agree with your opinion.
The following sentences were added to the Radiomics-based prediction of histological grades section.
These prediction models such as radiomics-based deep learning models have limitations including small sample size, class imbalance, overfitting risk, and the limited external or prospective validation. To avoid overinterpretation of the studies, understanding these limitations in the studies on outcome prediction models are required.
7. Terminology regarding ACT vs. grade 1 chondrosarcoma and WHO 2013 vs. 2020 classifications should be unified throughout the text, with a clear explanatory paragraph early in the introduction.
Response:
Thank you for your suggestion.
Terminology regarding ACT vs. grade 1 chondrosarcoma was unified throughout the manuscript. Throughout the manuscript, the term “ACT/grade 1 chondrosarcoma” is consistently used to refer to histologically low-grade cartilaginous tumors. Furthermore, explanation paragraph of ACT and grade 1 chondrosarcoma in the WHO 2013 and 2020 were added to the manuscript.
8. The suggestion that radiological grading might serve as an alternative to biopsy is currently too strong; it should be toned down and conditioned on future prospective, ideally multicenter, validation studies comparing combined histological and radiological approaches.
Response:
Thank you for your suggestion.
The sentence "improvements in reproducibility may enable radiological grading to be an alternative diagnostic tool for biopsies" was corrected as "use of radiological grading as an alternative tool for biopsies require prospective, multicenter validation".
9. The future perspectives section should better define specific research priorities (e.g., standardized imaging protocols, multicenter radiomics datasets, explainable AI) and how they could be translated into clinical decision algorithms.
Response:
Thank you for this suggestion.
In the Future perspectives and implications section, the following sentences were added.
Advances in standardized imaging protocols, multicenter radiomics datasets, and explainable AI are needed to apply these techniques into clinical decision algorithms.
10. The manuscript would benefit from clearer integration of the narrative: for each modality and each model, explicitly link how the reported diagnostic or prognostic performance could change concrete surgical decisions or follow-up strategies.
Response:
Thank you for your suggestion.
Histological diagnosis of tumor specimen is most important for surgical decision-making and follow-up strategies. However, high incidences of discrepancies of histological grade between biopsy specimen and resected tumor specimen have been reported. Enchondroma, ACT, and grade 1 chondrosarcoma can be treated with curettage, whereas grade 2 or higher chondrosarcoma should be resected with wide surgical margin. Patients with the histological discrepancies undergo inappropriate surgical treatment due to the low reliability of preoperative histological grade. Integrated diagnosis of preoperative histological and radiological assessment may improve the diagnostic accuracy, resulting in appropriate decision-making for surgical procedures. These sentences were added to the Future perspectives and implications section.
Round 2
Reviewer 3 Report
Comments and Suggestions for Authors
-
While new content has been added to the Future perspectives and clinical implications section for several points, the manuscript would benefit from slightly redistributing key clarifications into earlier relevant sections (e.g., radiomics methodology, scoring systems) to avoid overloading the future perspectives section.
-
In a few responses, the language could be edited for grammatical clarity and precision (for example, in the radiomics standardization and outcome prediction model sections) to improve readability and ensure a polished final version.
Author Response
1. While new content has been added to the Future perspectives and clinical implications section for several points, the manuscript would benefit from slightly redistributing key clarifications into earlier relevant sections (e.g., radiomics methodology, scoring systems) to avoid overloading the future perspectives section.
Response:
Thank you for your suggestion.
To avoid overloading the future perspectives section, the following sentences were redistributed to 2. Radiological findings to predict histological grade, 3. Comprehensive radiological assessment to predict histological grades, and 4. Radiomics-based prediction of histological grades sections.
2. Radiological findings to predict histological grade
Each modality provides complementary information, and comprehensive assessment of the modalities may improve diagnostic accuracy and make it possible to determine the appropriate surgical procedure. However, radiological grading has several limitations including interobserver variability, subjective interpretation, and modality-dependent differences. Radiological assessment by single imaging modality may have low reliability, and comprehensive and quantitative radiological assessment may prevent the misdiagnosis of grading in cartilaginous tumors.
3. Comprehensive radiological assessment to predict histological grades
In contrast, previous reports suggest that comprehensive assessments using several radiological findings with quantitative assessment could enable the evaluation of the aggressiveness of cartilaginous tumors.
4. Radiomics-based prediction of histological grades
Prediction models that use radiomics and machine learning allow the extraction of many quantitative features that reflect tumor aggressiveness. These techniques may enable the prediction of clinical outcomes, including metastasis and local recurrence. However, these prediction models such as radiomics-based deep learning models have limitations including small sample size, retrospective study, heterogeneity of imaging protocols, class imbalance, overfitting risk, and the limited external or prospective validation. Additionally, radiomics-based prediction models are sensitive to image acquisition and reconstruction parameters, which may affect reproducibility. Understanding these limitations in the studies on outcome prediction models are required to avoid overinterpretation of the studies. To apply radiomics-based prediction model into clinical practice, high reproducibility with technical standardization including acquisition parameters, segmentation methods, feature harmonization, and cross-scanner variability are developed and validated in prospective multicenter studies. Furthermore, advances in standardized imaging protocols, multicenter radiomics datasets, and explainable AI are needed to apply these techniques into clinical decision algorithms.
2. In a few responses, the language could be edited for grammatical clarity and precision (for example, in the radiomics standardization and outcome prediction model sections) to improve readability and ensure a polished final version.
Response:
Thank you for pointing this out.
The sentence “To apply radiomics-based prediction model into clinical practice, high reproducibility with technical standardization including acquisition parameters, segmentation methods, feature harmonization, and cross-scanner variability.” was corrected as follows.
To apply radiomics-based prediction model into clinical practice, high reproducibility with technical standardization including acquisition parameters, segmentation methods, feature harmonization, and cross-scanner variability are demanded.
The sentence “To avoid overinterpretation of the studies, understanding these limitations in the studies on outcome prediction models are required.” was corrected as follows.
Understanding these limitations in the studies on outcome prediction models is required to avoid overinterpretation of the studies.